# Are free school meals failing families? Exploring the relationship between child food insecurity, child mental health and free school meal status during COVID-19: national cross-sectional surveys

Tiffany C Yang [ORCID],[1] Madeleine Power [ORCID],[2] Rachael H Moss [ORCID],[1] Bridget Lockyer [ORCID],[1] Wendy Burton [ORCID],[2] Bob Doherty [ORCID],[3] Maria Bryant [ORCID] [2,4]

[1]Bradford Teaching Hospitals NHS Foundation Trust, Bradford, UK
[2]Health Sciences, University of York, York, UK
[3]The York Management School, University of York, York, UK
[4]York Hull Medical School, University of York, York, UK

**Correspondence to**
Dr Tiffany C Yang;
tiffany.yang@bthft.nhs.uk

## ABSTRACT

**Objective** Food insecurity is linked to poor health and well-being in children and rising prevalence rates have been exacerbated by COVID-19. Free school meals (FSM) are considered a critical tool for reducing the adverse effects of poverty but apply a highly restrictive eligibility criteria. This study examined levels of food security and FSM status to support decision-making regarding increasing the current eligibility criteria.

**Design** Two cross-sectional national surveys administered in August–September 2020 and January–February 2021 were used to examine the impact of COVID-19 on the food experiences of children and young people.

**Setting** UK.

**Participants** 2166 children (aged 7–17 years) and parents/guardians.

**Main outcome measures** Participant characteristics were described by food security and FSM status; estimated marginal means were calculated to obtain the probability of poor mental health, expressed as children reporting feeling stressed or worried in the past month, by food security status and FSM status.

**Results** We observed food insecurity among both children who did and did not receive of FSM: 23% of children not receiving FSM were food insecure. Children who were food insecure had a higher probability of poor mental health (31%, 95% CI: 23%, 40%) than children who were food secure (10%, 95% CI: 7%, 14%). Food insecure children receiving FSM had a higher probability of poor mental health (51%, 95% CI: 37%, 65%) than those who were food insecure and not receiving FSM (29%, 95% CI: 19%, 42%).

**Conclusion** Many children experienced food insecurity regardless of whether they received FSM, suggesting the eligibility criteria needs to be widened to prevent overlooking those in need.

## INTRODUCTION

The past decade has witnessed sharp rises in the use of food banks by households with children, suggesting that child food insecurity is rising.[1] In that decade, the Trussell Trust reported a 31-fold increase in the number of emergency food parcels distributed, from 61 000 in 2010–2011 to 1.9 million in 2019/2020.[2] Food insecurity can broadly be defined as uncertainty around the quality and quantity of food available.[3] Data from the Family Resource Survey (FRS) prior to the COVID-19 pandemic indicates that households with children are at particular risk of food insecurity in the UK.[3–5] In 2019/2020, five million people in the UK (8%) were in food insecure households, of whom 13% were children, 8% were working age adults and 2% were pensioners.[4] Food insecurity has considerable nutritional, physical and cognitive implications for children including, but not limited to, associations with lower vegetable intake, higher added sugar intake,[6 7] increased risk of obesity[5 8–10] and poorer academic performance.[11 12] There is a growing body of literature—almost entirely

from North America—evidencing an association between the experience of food insecurity and an increase in the risk of mental health issues for children and adolescents.[13–18] Children and teenagers experiencing food insecurity report lower life satisfaction,[14] and have a higher probability of seeing a psychologist and finding it difficult to make friends.[19] Evidence suggests that rates of depression,[20] stress and anxiety are higher for children living in food insecure households.[15 21 22] Households with children have been particularly badly affected by the social and economic fallout of the COVID-19 pandemic. In the first 6 months of the pandemic, 12% of adults living with children reported skipping meals because they could not afford or access food, while 4% of adults with children reported going for a whole day without eating.[23] Food banks also reported a sharp rise in access by households with young children. Between early and mid-2020, The Trussell Trust food bank network supported 370 000 households, of which 320 000 were families with children. The proportion of couples with children referred to a food bank increased from 19% in early 2020 to 24% during the COVID-19 pandemic in mid-2020.[2]

Free school meals (FSM) are considered to be a critical tool for mitigating the negative health effects of child poverty among low-income families. Children receiving FSM obtain a higher proportion of their daily energy and nutrient intakes from their school meals compared with those who pay[24 25] and FSM may therefore improve health and well-being and reduce health inequalities.[26 27] In England, FSM are currently a statutory entitlement available to eligible pupils, which include all infant school children (reception, year 1 and year 2) in state-funded schools (as part of the Education Act, 1944)[28]; and pupils in year three and upwards (junior school and secondary school pupils) whose parents meet income-defined eligibility criteria (parents currently meet the eligibility criteria if they receive: Income Support; Income-based Jobseekers Allowance; Income-related Employment and Support Allowance; Support under Part 5 of the Immigration and Asylum Act 1999; the guaranteed element of State Pension Credit; Child Tax Credit (provided they are not also entitled to Working Tax Credit and have an annual gross income of no more than £16 190); Working Tax Credit run-on (paid for 4 weeks after a person stops qualifying for Working Tax Credit); and Universal Credit (with household income of less than £7400 a year after tax and not including any benefits)) and claim for FSM. As of 1 October 2020, there were 1.63 million pupils known to be eligible for FSM, including those part of the universal FSM offer,[29] an increase in the proportion eligible to 19.7% of all state-funded pupils from 17.3% in January 2020 to 15.4% in January 2019. This increase is likely due to increased unemployment during the COVID-19 pandemic rendering more children eligible for FSM, alongside increased uptake due to greater media attention and awareness of FSM.

FSM receipt can be considered a marker of poverty due to its restrictive eligibility criteria and children who receive FSM are likely to be living in low-income households. The COVID-19 pandemic has exposed and amplified pre-existing concerns about the restrictive eligibility criteria for FSM (for pupils above year 2) and low uptake of FSM among eligible families (both before and after registration).[30] The effects of the pandemic have been highly unequal, according to income, ethnicity, gender and health status.[31–37] There is evidence to suggest that low-income families have been particularly negatively affected by the social and economic circumstances of the pandemic,[38–40] and yet have thus far been largely neglected in the Governmental policy response. Emerging evidence suggests that families just outside of the eligibility criteria for FSM have struggled to afford food during the pandemic, potentially exacerbating child food insecurity.[41] However, this has not yet been formally assessed. This paper addresses this important and urgent research gap, examining the relationship between child food insecurity and families who did or did not receive of FSM during the pandemic. Given the known negative effects of food insecurity on child mental health and educational outcomes,[11 13 42] the paper also looks at child mental health in the context of child food insecurity and receipt of FSM.

## METHODS
### Study population and survey design
Data were taken from two Food Foundation commissioned surveys (ChildWise) conducted in the course of the COVID-19 pandemic to examine its impact on children and young people's COVID-19 food experiences. The first (August–September 2020; response rate: 10%) and second surveys (January–February 2021; response rate: 28%) were carried out online using a carefully constructed framework to ensure a geographic and demographic representative sample of adults living in the UK with children and young people aged 7–17 years. Children younger than 7 years of age, including children aged 7 in year 2 of primary school, were excluded in order to capture children's experiences outside of universal FSM provision.

The online panel used by ChildWise is a member of the European Society of Opinion and Marketing Research organisation and endeavours to be as representative as possible. This panel is the largest in the UK and globally. To achieve representative quotas, the panel's profiling data were first used to target the more difficult-to-reach demographics before targeting other groups. Samples were recruited to be representative by region, broad ethnic group and spread evenly by age and gender.

Surveys were completed by parents or guardians (hereafter 'parents') of children with a section to be completed by children with the aid of parents if required. Parents were asked to list the ages and genders of all children in the household and one child was initially randomly allocated to complete the child portion of the survey. Parents completed questions on sociodemographic details and

**Table 1** Characteristics of the survey population

| | Total sample n=2166 | |
|---|---|---|
| | **N** | **Mean (SD)/%** |
| **Parent responses** | | |
| **Parent age** | | |
| 18–24 | 8 | 0.4 |
| 25–34 | 268 | 12.4 |
| 35–44 | 923 | 42.6 |
| 45–54 | 762 | 35.2 |
| 55–64 | 205 | 9.5 |
| Missing | – | – |
| **Parent occupation** | | |
| Higher | 1341 | 61.9 |
| Lower | 825 | 38.1 |
| Missing | – | – |
| **Geographical region** | | |
| East Midlands | 158 | 7.3 |
| Eastern | 196 | 9 |
| London | 282 | 13 |
| North East | 92 | 4.2 |
| North West | 240 | 11.1 |
| Northern Ireland | 73 | 3.4 |
| Scotland | 161 | 7.4 |
| South East | 300 | 13.9 |
| South West | 197 | 9.1 |
| Wales | 109 | 5 |
| West Midlands | 182 | 8.4 |
| Yorkshire and Humberside | 176 | 8.1 |
| Missing | – | – |
| **Number in household** | | |
| 2 | 160 | 7.4 |
| 3 | 624 | 28.8 |
| 4 | 939 | 43.4 |
| 5 | 318 | 14.7 |
| 6+ | 125 | 5.8 |
| Missing | – | – |
| **Child ethnicity** | | |
| Asian | 245 | 11.4 |
| Other* | 209 | 9.7 |
| White | 1691 | 78.8 |
| Missing | 21 | – |
| **Child age (years)** | 2166 | 12.4 (3.2) |
| Missing | – | – |
| **Child sex** | | |
| Female | 1076 | 49.7 |
| Male | 1090 | 50.3 |
| Missing | – | – |

Continued

**Table 1** Continued

| | Total sample n=2166 | |
|---|---|---|
| | **N** | **Mean (SD)/%** |
| **Child receives FSM** | | |
| Yes | 675 | 31.5 |
| No | 1467 | 68.5 |
| Missing | 24 | – |
| **Child responses** | | |
| **Potential food insecurity** | | |
| Yes | 431 | 20.6 |
| No | 1659 | 79.4 |
| Missing | 76 | – |
| **Any food bank use** | | |
| Yes | 561 | 25.9 |
| No | 1605 | 74.1 |
| Missing | – | – |
| **Food insecure†** | | |
| Yes | 763 | 35.2 |
| No | 1403 | 64.8 |
| Missing | – | – |
| **Find FSM embarrassing** | | |
| Yes | 62 | 9.7 |
| No | 578 | 90.3 |
| Missing | 1526 | – |
| **Stressed/worried‡** | | |
| Every/most days | 236 | 18 |
| Some/rarely | 1053 | 82 |
| Missing | 19 | – |

*The other ethnicity category includes the following groups: black African, black Caribbean, other black background, mixed and other background.
†Defined as responding affirmatively to any of the six potential food insecurity questions or indicated any food bank use.
‡Responses available only among a children participating in the January–February 2021 survey.
FSM, free school meals.

were asked to complete information about up to two of their children's FSM status, age and gender. Children completed questions on perception of FSM, food insecurity and food bank use. In the second survey, additional questions on the child's mental health were included in the children's section. Towards the end of the fieldwork period, children were non-randomly assigned to complete the child portion of the survey based on fulfilling any remaining quotas of age, gender and geographic region.

## Patient and public involvement

The survey used in this study was developed in partnership with Food Foundation, who have established a group of young food ambassadors to help set priority areas. This group meets on a regular basis to discuss important and

**Table 2** Food insecurity and food bank use by children who receive or do not receive free school meals

| | Received FSM | | | | | Did not receive FSM | | | | |
|---|---|---|---|---|---|---|---|---|---|---|
| | Food insecurity n=407 (60%) | | No food insecurity n=268 (40%) | | | Food insecurity n=338 (23%) | | No food insecurity n=1129 (77%) | | |
| | N | Mean (SD)/% | N | Mean (SD)/% | P value* | N | Mean (SD)/% | N | Mean (SD)/% | P value* |
| **Parent age** | | | | | <0.001 | | | | | <0.001 |
| 18–24 | 5 | 1.2 | 0 | 0 | | 1 | 0.3 | 2 | 0.2 | |
| 25–34 | 91 | 22.4 | 44 | 16.4 | | 43 | 12.7 | 87 | 7.7 | |
| 35–44 | 194 | 47.7 | 115 | 42.9 | | 167 | 49.4 | 436 | 38.6 | |
| 45–54 | 97 | 23.8 | 75 | 28 | | 107 | 31.7 | 474 | 41.9 | |
| 55–64 | 20 | 4.9 | 34 | 12.7 | | 20 | 5.9 | 130 | 11.5 | |
| **Parent occupation** | | | | | <0.001 | | | | | 0.1 |
| Higher | 238 | 58 | 106 | 40 | | 214 | 63 | 765 | 68 | |
| Lower | 169 | 42 | 162 | 60 | | 124 | 37 | 364 | 32 | |
| **Number in household** | | | | | 0.6 | | | | | 0.2 |
| 2 | 37 | 9.1 | 29 | 10.8 | | 25 | 7.4 | 63 | 5.6 | |
| 3 | 123 | 30.2 | 80 | 29.9 | | 106 | 31.4 | 311 | 27.5 | |
| 4 | 164 | 40.3 | 95 | 35.4 | | 142 | 42 | 527 | 46.7 | |
| 5 | 51 | 12.5 | 43 | 16 | | 46 | 13.6 | 177 | 15.7 | |
| 6+ | 32 | 7.9 | 21 | 7.8 | | 19 | 5.6 | 51 | 4.5 | |
| **Child ethnicity** | | | | | 0.04 | | | | | 0.8 |
| Asian | 66 | 16 | 30 | 11 | | 35 | 10.5 | 110 | 9.8 | |
| Other | 57 | 14 | 28 | 10 | | 30 | 9 | 91 | 8.1 | |
| White | 282 | 70 | 210 | 78 | | 268 | 80.5 | 916 | 82 | |
| **Child age** | 407 | 11.9 (3.1) | 268 | 12.2 (3.2) | 0.2 | 338 | 11.5 (3.1) | 1129 | 12.9 (3.1) | <0.0001 |
| **Child sex** | | | | | 0.6 | | | | | 0.9 |
| Female | 182 | 45 | 126 | 47 | | 176 | 52 | 580 | 51 | |
| Male | 225 | 55 | 142 | 53 | | 162 | 48 | 549 | 49 | |
| **Child finds FSM embarrassing†** | | | | | <0.001 | | | | | |
| No | 313 | 86.9 | 233 | 95.5 | | – | – | – | – | |
| Yes | 47 | 13.1 | 11 | 4.5 | | – | – | – | – | |
| **Child stressed/worried** | | | | | <0.001 | | | | | <0.001 |
| Every/most days | 102 | 41 | 17 | 10 | | 43 | 22 | 68 | 10 | |
| Some/rarely | 144 | 59 | 145 | 90 | | 152 | 78 | 605 | 90 | |

*$\chi^2$, Fisher's exact or Welch's two-sample t-test.
†Only children responding affirmatively to receiving FSM were asked about this item.
FSM, free school meals.

emerging areas of interest that have the best chance of policy change. They have provided advice on asking questions to young people and the methodological approach used.

### Sociodemographic characteristics

Parents completed questions on their and their child's age and gender, their child's ethnicity and occupation of the Chief Income Earner. We collapsed ethnicity from 12 categories into three: white (white British; other white background); Asian (Bangladeshi; Chinese; Indian; Pakistani; other Asian background) and other (black African; Black Caribbean; other black background; mixed background). Participants who chose 'prefer not to answer' were coded as missing.

Parental occupation was reported for the Chief Income Earner in the household, defined as the individual within the household with the largest income. If the Chief Income Earner was not in paid employment but has been out of work for fewer than 6 months, the most recent occupation was reported. If the Chief Income Earner was retired with an occupation pension, then the most recent occupation was reported. Twelve categories

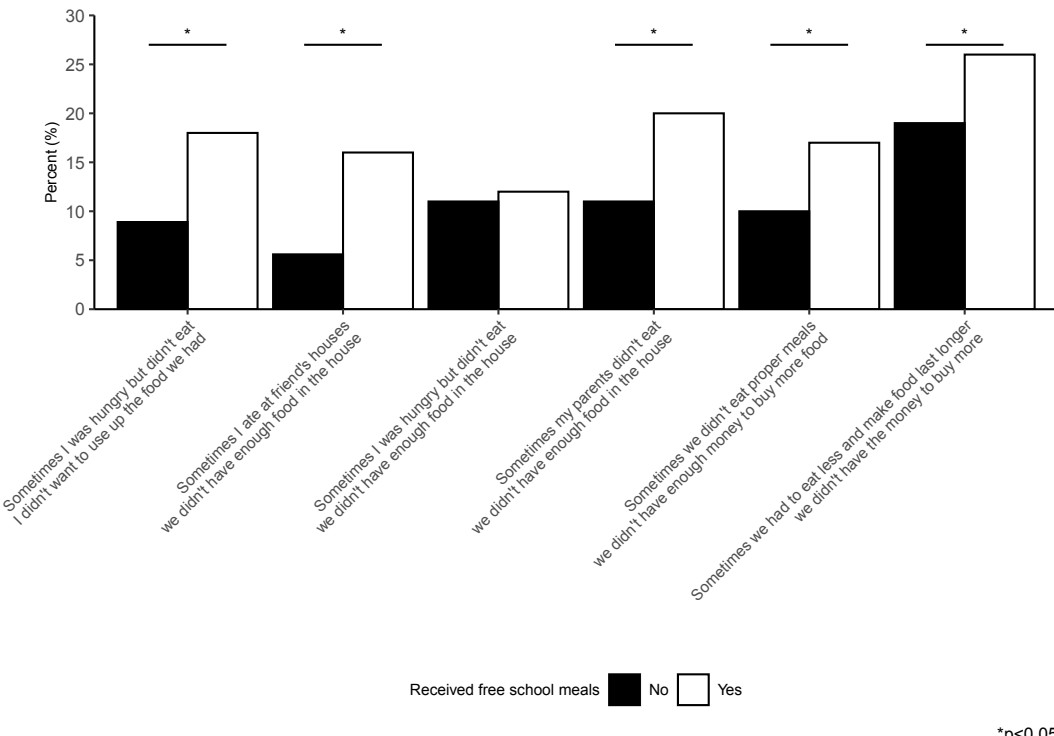

*p<0.05

**Figure 1** Percentage of children who responded affirmatively to the six questions indicating potential food insecurity by free school meals status.

of occupations were collapsed by ChildWise into two categories of social grade with the higher occupational class as a shorthand for middle class (ABC1) and the lower occupational class as shorthand for working class (C2DE): Higher (Supervisory or clerical/ junior managerial/ professional/ administrative; Intermediate managerial/ professional/ administrative; Higher managerial/ professional/ administrative; Student) and Lower (Semi or unskilled manual work; Skilled manual worker; Casual worker - not in permanent employment; Housewife/ Homemaker; Retired and living on state pension; Unemployed or not working due to long-term sickness; Full-time carer of other household member; Other).

### Free school meals

Parents were asked whether their child was currently registered for FSM. Responses were coded to 'yes' if parents responded 'yes' and 'no' if parents responded 'no'. Responses of 'don't know' and 'prefer not to say' were coded as missing. Responses were similar to the question asked of children (thinking about when you have lunch at school, do you have free school meals?).

### Food insecurity

Children were asked to think about being at home during the summer holidays (August–September 2020 survey) or the Christmas holidays and recent lockdown (January–February 2021 survey) and asked to respond to several statements regarding potential food insecurity. Children were categorised as having 'potential food insecurity'

if they responded 'yes' to any of the following six statements: (1) sometimes I was hungry but didn't eat because I didn't want to use up the food we had; (2) sometimes I was hungry but didn't eat because we didn't have enough food in the house; (3) sometimes my parents didn't eat because we didn't have enough food in the house; (4) sometimes we had to eat less and make food last longer because we didn't have the money to buy more; (5) sometimes we didn't eat proper meals because we didn't have enough money to buy more food; and (6) sometimes I ate at friend's houses because we didn't have enough food in the house. Children were categorised as not having potential food insecurity if they responded 'yes' to 'I always had enough food to eat'. There were no children who responded affirmatively to both 'I always had enough to eat' and any of the other six statements. Children who responded affirmatively to 'don't know' or 'prefer not to say' were coded as missing.

Children were also asked to respond to several statements regarding food bank use. They were coded to any food bank use if they responded 'yes' to having visited a food bank by themselves or if their family visited or if they responded 'no' to the statement 'no, we didn't visit a food bank'. Remaining children were coded as not having used a food bank.

A dichotomous variable of 'food insecurity' was then generated and included children who were identified as having 'potential food insecurity' (from the six questions) or indicated any food bank use. Children who did

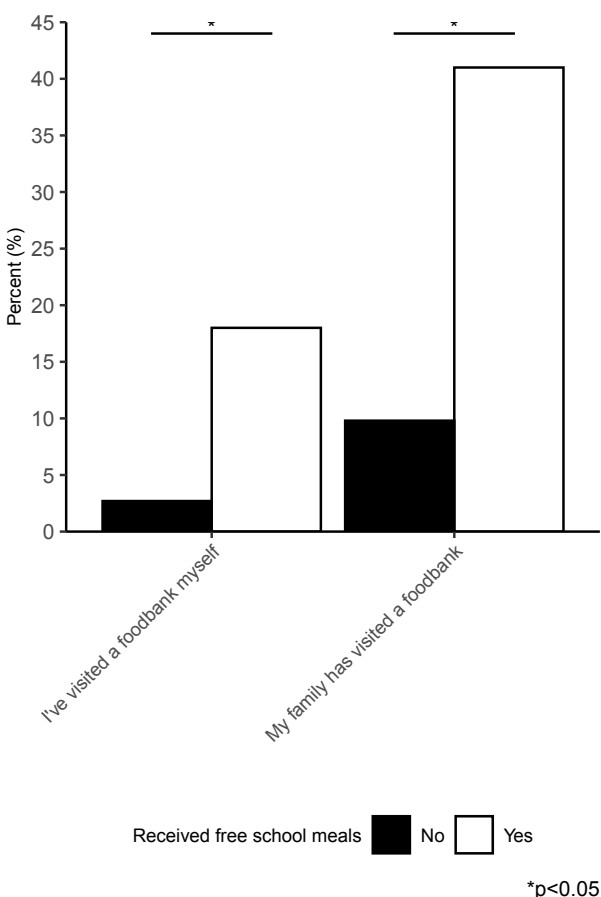

**Figure 2** Percentage of children responding affirmatively to two questions indicating food bank use by free school meals status.

not have 'potential food insecurity' and did not indicate any food bank use were considered to be 'food secure'. Children who responded 'yes' to any of the 'potential food insecurity' questions or indicated that they or their family had visited a food bank were considered to be 'food insecure'.

### Mental health

Among children who responded affirmatively that they received FSM, they were asked to select from a range of statements on how they felt about FSM. We examined their responses to 'I think it is embarrassing to have free school meals'; affirmative responses were coded to 'yes, embarrassed' and negative responses were coded to 'no, not embarrassed'.

Children participating in the January–February 2021 survey were asked how often they felt stressed or worried in the past month and were categorised as being stressed or worried 'every/most days' if they said they were worried 'every day' or 'most days'. Children were categorised as being stressed or worried 'some/rarely' if they said they were worried 'some days', 'rarely' or 'I have not felt stressed once in the last month'.

### Data analysis

All analyses were conducted in R V.4.0.2.[43] Responses were combined and analysed across both surveys. We examined differences in characteristics by survey period and did not find differences by measures of food insecurity or receipt of FSM (online supplemental file 1). A small number of participants responded to both surveys (n=206). Their responses were removed from the first survey so that they were present in the sample only once and were able to be part of the mental health analysis. Participants who responded to both surveys were less likely to have used a food bank or be food insecure (online supplemental file 2). Main analyses were completed on a sample size of n=2166.

Responses were described using mean (SD) for continuous measures and number (n) and percentage (%) for categorical measures. We constructed four groups: (1) children with food insecurity who received FSM; (2) children with food insecurity who did not receive FSM; (3) children without food insecurity who received FSM; and (4) children without food insecurity who did not receive FSM.

Differences between participant characteristics and responses to food insecurity questions, food bank use, and derived food insecurity by children who received or did not receive FSM were assessed using $\chi^2$ or Fisher's exact tests for categorical variables and Welch's two-sample t-test for continuous variables. A significant p value (p<0.05) indicates that there is a difference between the characteristics by food security status among children who received FSM and among children who did not receive FSM. In the subset of survey questions on children's mental health, unadjusted and adjusted logistic regression using complete case analysis were run and estimated marginal means obtained. Fully adjusted analyses were performed with n=1265 participants. A directed acyclic diagram was drawn to assist in the selection of covariates (online supplemental file 3). In fully adjusted analyses of food insecurity with child mental health, we included child age and sex, child ethnicity, parent occupation, household occupancy, region, receipt of FSM and an interaction term between food insecurity and FSM. We calculated the probability of our outcome for every combination of food security status and FSM status while holding all covariates at their mean or mode using the 'predictions' function in the 'marginal effects' package.

### RESULTS
### Participant characteristics
The majority (77.8%) of parent respondents were aged 35–54 years old, were professionally employed (61.9%) and lived in households with three or four people (72.2%) (table 1). The majority of children were white (78.8%) or Asian (11.4%) and just under a third of parents reported that their child received FSM (31.5%). Over 20% of children reported potential food insecurity, based on positively responding to any measure, and over a quarter

reported that they or their family had visited a food bank (25.9%), placing over a third of children living with food insecurity according to our definition (35.2%). Among children who affirmed that they received FSM, a tenth reported that receiving FSM is embarrassing. Almost a fifth (18%) of children responding to the January–February 2021 survey reported that they felt stressed or worried every day or most days.

Among children who received FSM, 60% were considered to have food insecurity (table 2). Parents of children receiving FSM were younger, were more likely to be in a lower level of occupation and less likely to be of white ethnicity. Over 20% of children who did not receive FSM reported food insecurity. Parents of children who did not receive FSM and had food insecurity were more likely to be younger than those not living with food insecurity. Among children who did not receive FSM, there was no difference in parental occupation between those with lived with or without food insecurity, with parents in both groups more likely to have a higher level of occupation. Children who were food insecure and and who received FSM were more likely to express that receiving FSM is embarrassing (13.1%) compared with those who did not receive FSM (4.5%; p<0.001). Children experiencing food insecurity were more likely to report feeling stress or worried every day or most days and this was greater among children who received FSM than not.

### FSM and potential food insecurity
Children who received FSM were more likely to have reported any potential food insecurity measure (42.8%) compared to those who did not receive FSM (9.8%; p<0.05). Figure 1 shows the percentage of children who responded affirmatively to each of the six potential food insecurity questions by FSM status. Children who received FSM were more likely to respond affirmatively to these questions, though many who did not receive FSM also indicated potential food insecurity. Among all children, the most commonly chosen item reported was having to eat less in order to make food last longer due to a lack of money to buy more food.

### FSM and food bank use
Both children who did and did not receive FSM reported visiting a food bank, whether by themselves or their family (figure 2). Children who received FSM were more likely to have reported visiting a food bank by themselves or their family than children not receiving FSM. We found 2.7% of children who did not receive FSM visited a food bank by themselves, while 9.8% reported their families had visited a foodbank.

### Child's mental health
The probability of a child reporting being stressed or worried every day or most days was 31% (95% CI: 23%, 40%) among those reporting food insecurity and 10% (95% CI: 7%, 14%) among those not reporting food insecurity, adjusting for confounders (figure 3). In models

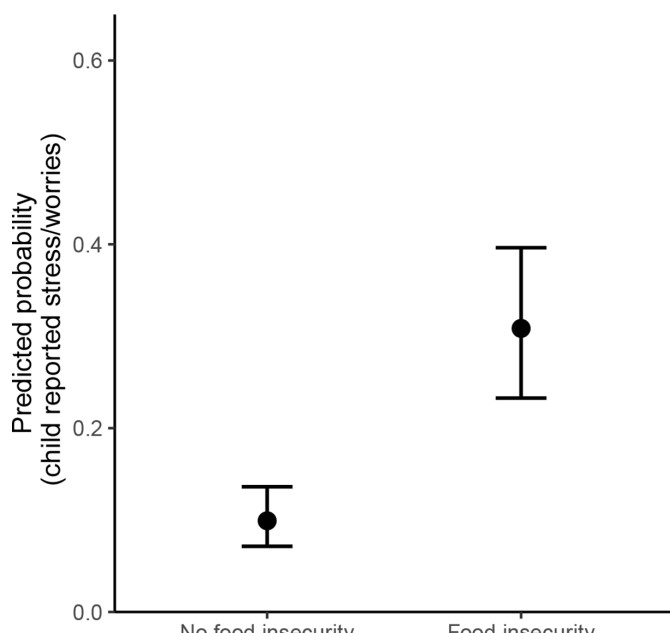

**Figure 3** Probability of a child reporting feeling stressed or worried every day or most days in the past month by food security status.

additionally examining FSM, the probability of a child reporting being stressed or worried every day was 51% (95% CI: 37%, 65%) among children with food insecurity and in receipt of FSM (figure 4). Among children with food insecurity but not in receipt of FSM, the probability was 29% (95% CI: 19%, 42%).

### DISCUSSION
In this family-based survey measuring experiences of the COVID-19 pandemic, we found a substantial number of children experienced food insecurity (defined here as having ever visited a food bank or experienced any food insecurity measure) regardless of whether they received FSM. Food insecurity and measures of potential food insecurity were highest among children who received FSM, likely reflecting the very low-income threshold for FSM, meaning that outside of universal infant provision (in England and Scotland), it is largely children in the very poorest families who receive FSM. In a subset of children with mental health measures, we found that children who experienced food insecurity were more likely to report feeling stressed or worried on an almost-daily basis in the previous month compared with children who were food secure.

Children are often protected from hunger in families that experience food insecurity as parents report decreasing their own intake to shield their children.[44–47] We found that children reported their parents skipping meals due to a lack of food in the house. However, it was concerning that we also found children reporting hunger due to not having enough food in the house and almost 20% of all children living with food insecurity reporting having to eat less to make food stretch. The high

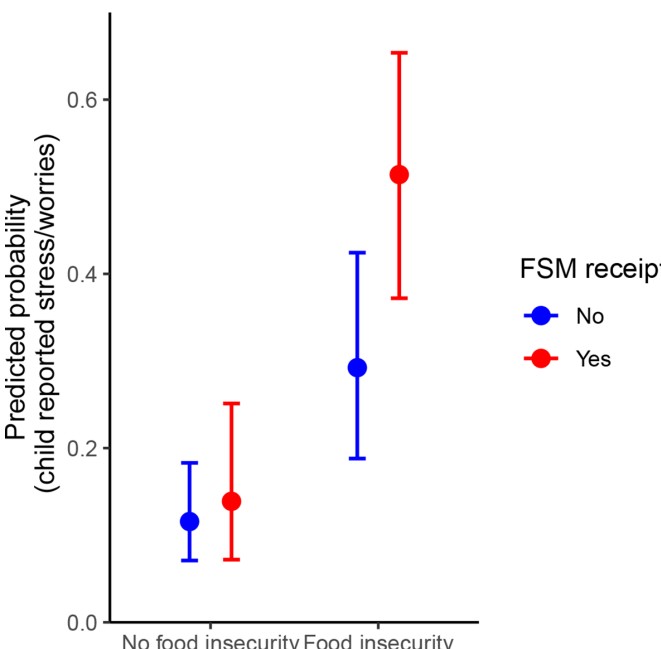

**Figure 4** Probability of a child reporting feeling stressed or worried every day or most days in the past month by food security and free school meals status.

proportion of children with food insecurity reporting food bank use is consistent with other reports that have highlighted the impact of the pandemic on levels of food insecurity.[1] The consequences of the pandemic on financial stability and, therefore, food insecurity have also impacted families who may not have been previously affected.

Studies have observed food insecurity among the employed and recent data in the UK Longitudinal Household Study on food insecurity during the pandemic suggest that while risk of food insecurity increased more for those who were unemployed, those who were persistently employed were also at risk.[48–51] The pandemic has exposed the notion that food insecurity occurs primarily among the unemployed or those less-skilled professions; over 50% of children reporting food insecurity in our data had parents with higher/professional levels of occupation. Educational attainment and income are not necessarily protective of food insecurity. In the 2019/2020 FRS, it was reported that 8% of households where the head of household obtained A-levels or Scottish Highers and 4% of households with further education and university qualifications were food insecure.[4] Likewise, increasing income increased food security but even among those with a gross weekly income of £1000 or more only 96% had high food security (meaning 4% experienced food insecurity), while only 74% of households with a total gross weekly income of less than £200 had high food security.

Over 25% of all children and over 50% of children living with potential food insecurity reported their families visiting a food bank. Previous research has suggested that use of food banks by UK households experiencing

food insecurity is low[52 53]; however, our study suggests that in the context of the pandemic, food bank use may have become more common for families experiencing food insecurity. Food banks are a short-term 'emergency' response and concerns have been raised about the nutritional quality[54 55] and cultural adequacy of the food provided.[56] The emergence and continuation of food banks and the growing number of food parcels they provide may be seen as an example of 'successful' self-organisation around a need and conveys a sense that something is being done; however, it should be questioned whether it is the responsibility of the voluntary sector, rather than the Government, to provide access to something as basic as having food of sufficient quality and quantity. Food banks are often unable to provide fresh foods or ensure dietary requirements are met; continued reliance and widespread use of food banks, particularly among households with children, raises concerns about the long-term mental and physical implications for families relying on this form of 'emergency' support.

FSM are often seen as an essential tool for mitigating the effects of poverty experienced by children but provision is not universal or standardised across all nations, leading to unequal access[41]. Our data reported more children registered for FSM (32%) compared with those reported eligible in autumn 2020 (19.7%).[29] This may be reflected by an increase in newly qualifying children for FSM as families lost income; in the first survey, over 40% of children registered for FSM had only recently started receiving FSM (ie, were newly eligible that term).[23] Once children age out of universal provision, stringent criteria must be met for children to receive FSM with many of the criteria being income based, leading to only very low-income families being eligible and many low-income families going hungry. The eligibility is so restrictive that in our sample nearly half of families who are food insecure do not receive FSM; the eligibility threshold is set at an annual household income of less than £7400 prior to benefits, while parents receiving Working Tax Credit are ineligible for FSM support regardless of their level of income. However, as we have shown, a large proportion of children experiencing food insecurity as well as those in receipt of FSM have parents employed in professional-level occupations. This suggests that the financial circumstances of families of all income levels have been hard hit by the pandemic and that the current criteria may not be suitable for assessing eligibility. In addition, our findings that children who were receiving FSM still reported hunger suggests that FSM provision may not be sufficient to ensure that children are adequately fed on a daily basis.

There is limited published research in the UK on the role food insecurity plays in children's mental health, and none on the role of FSM in mitigating the association between food insecurity and poor mental health among children. Emerging UK evidence suggests poorer well-being and increased emotional and behavioural problems among children who experience food insecurity. One UK study found 27% of 10-year-old children experiencing

food insecurity exhibited clinically significant behavioural problems compared with 10% of children who were food secure.[57] Our findings that children who experience food insecurity have worse mental health are therefore unsurprising and in line with North American literature on food insecurity and child mental health.[13–18] We found that children who reported food insecurity and received FSM had a higher probability of reporting feeling stressed or worried compared with children who did not receive FSM. This could potentially reflect the complex poverty-related stressors of living in a household eligible for FSM, and could indicate a more severe level of socioeconomic deprivation among children reporting food insecurity and receiving FSM, as well as the perceived stigma of receiving FSM.[38 58] Children who received FSM in this survey were asked whether they think it is embarrassing to have FSM and 13.1% of children living with food insecurity thought it was embarrassing compared with 4.5% of children who were food secure. This suggests children may carry an additional burden of stigma on top of inadequate food security.

There are multiple strengths and limitations to this study. This study used a geographic and demographically representative sample with a wide range of ages and included measures of child-reported food insecurity combined with child reported mental health. However, we were unable to understand the degree to which parents may have helped their child complete the questions, and whether responses were given by them or were changed/given by their parents. This is more likely to have influenced responses from the younger children completing the survey. While the online panel used by ChildWise aims to be as representative as possible across geographic and demographic characteristics, it is possible that families within these representative categories who were more interested in the scope of the survey or who were food insecure would have participated, potentially skewing the responses towards those who experienced food insecurity or received FSM. Other studies conducted during the COVID-19 pandemic have also reported high prevalence of food insecurity, with one study finding 16.2% of adults surveyed during the first lockdown reporting food insecurity while the Understanding Society COVID-19 longitudinal study survey found the prevalence of food insecurity rose from 7.1% in April 2020 to 20.2% by July 2020.[50 51] We also combined responses across the two surveys and observed that there were fewer parents in the higher occupation category at the second survey (60%) compared with the first survey (65%) but did not find any differences in food insecurity, food bank use or FSM when we examined participant characteristics by survey, suggesting it was appropriate to combine surveys. As some respondents participated in both surveys, we removed them from the August–September 2020 survey and included them in the January–February 2021 survey in order to maximise sample size for the mental health analyses; we examined whether participants who responded to both surveys were different from those responding to only

the August–September 2020 survey and found that they were less likely to have visited a food bank or report food insecurity. Removing these participants from the analyses did not alter our results. We were unable to explore experiences by more granular categories of ethnicity given small sample sizes among, for example, children of black ethnicity, preventing examination of how other prevalent ethnic groups experience food insecurity. Similarly, categorisation of occupation into two groups may mask differences between occupations within each group and we only had occupation data for the Chief Income Earner, which may not adequately reflect the socioeconomic position of the household. We were additionally unable to examine the experiences of children who identify outside of the male–female binary as this information was not collected. The questions used to assess food insecurity and child's mental health were not from a standardised tool and therefore have not been validated; however, these questions still provide insight into the disruption of quality or quantity of food available and state of mind. We also did not stratify the sample to distinguish between children in primary or secondary school, which may reflect differences in how food insecurity is experienced, such as visiting a food bank themselves, as well as differential uptake in FSM regardless of eligibility. We were also unable to differentiate between eligibility and uptake and whether the 23% of children experiencing food insecurity but not receiving FSM were due to non-eligibility or from voluntary refusal as a result of stigma or other reasons for not participating when eligible, such as navigating the application process. Finally, due to the lack of information about parental income and other household financial constraints/resources, it is important to acknowledge that other than food insecurity and eligibility for FSM, we have not been able to identify any other factors (perhaps correlated with food insecurity) that may be impacting on a child's mental health. Future research should consider other factors such as parental income, household income and receipt of benefits to help provide a more descriptive and causal picture of the financial status of participants' households.

Our findings confirm a real need to reconsider the eligibility criteria currently set for the provision of FSM. A concerning number of children are experiencing food insecurity in families with higher/professional levels of education who are likely to be above the eligibility threshold for FSM. While more families can be helped by widening eligibility, and more work is needed to understand access and uptake of FSM, including potential barriers such as social shame, policies which provide universal coverage should be considered as the impact goes beyond food provision and eliminates the stigma that is associated with being eligible and receiving FSM.

**Acknowledgements** We would like to take this opportunity to thank The Food Foundation and ChildWise for allowing us to conduct analyses on their COVID-19

survey data and to the children and families who took part in the surveys. We thank Dr Brian Kelly for his assistance in the statistical analysis.

**Contributors**  TCY, MP, RHM, BL, WB, BD, and MB designed and planned the study. MB acquired the data. TCY performed the statistical analysis and was responsible for the initial draft of the manuscript. All authors were involved in interpreting the study results, revising the manuscript and approving the final version for submission. MB is the guarantor. The corresponding author attests that all listed authors meet authorship criteria and that no others meeting the criteria have been omitted.

**Funding**  This work is supported in part by the FixOurFood programme (BB/V004581/1) funded by the UK Research and Innovation Transforming Food Systems Programme; ActEarly UK Prevention Research Partnership Consortium (MR/S037527/1); The National Institute for Health Research under its Applied Research Collaboration Yorkshire and Humber (NIHR200166); and The Health Foundation COVID-19 Award (2301201).

**Competing interests**  None declared.

**Patient and public involvement**  Patients and/or the public were not involved in the design, or conduct, or reporting, or dissemination plans of this research.

**Patient consent for publication**  Not applicable.

**Ethics approval**  We performed a secondary analysis of data that were collected externally by ChildWise (commissioned by the Food Foundation). We have discussed the approach taken by ChildWise extensively, who explained that ethical approval is only sought if they feel the questions or topic area are of a sensitive or personal nature. ChildWise surveys have 'topic disclaimers' at the start and, where appropriate, respondents are provided with a list of helplines they could reach out to, and potentially sensitive questions include a 'prefer not to say' option. When the research was commissioned it was not a pre-existing requirement from the client commissioning ChildWise. Informed consent was not obtained among child participants as parental participation in the survey and their agreement that a child within their household would complete the child's portion of the survey was taken as assent. Data from this national survey are not identifiable.

**Provenance and peer review**  Not commissioned; externally peer reviewed.

**Data availability statement**  Data may be obtained from a third party and are not publicly available. The datasets analysed during the current study are not freely available. Applications to access the data can be made to The Food Foundation.

**ORCID iDs**
Tiffany C Yang http://orcid.org/0000-0003-4549-7850
Madeleine Power http://orcid.org/0000-0002-9571-1782
Rachael H Moss http://orcid.org/0000-0001-6160-8536
Bridget Lockyer http://orcid.org/0000-0002-2195-5549
Wendy Burton http://orcid.org/0000-0001-7885-5971
Bob Doherty http://orcid.org/0000-0001-6724-7065
Maria Bryant http://orcid.org/0000-0001-7690-4098

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
