## [Reviewer comments · BMJ Open]

ARTICLE DETAILS

TITLE (PROVISIONAL)	Are Free School Meals failing families? Exploring the relationship between child food insecurity, child mental health and Free School Meal status during COVID-19: national cross-sectional surveys
AUTHORS	Yang, Tiffany; Power, Madeleine; Moss, Rachael; Lockyer, Bridget; Burton, Wendy; Doherty, Bob; Bryant, Maria

VERSION 1 – REVIEW

REVIEWER	Moula, Zoe Imperial College London, Public Health
REVIEW RETURNED	01-Dec-2021

GENERAL COMMENT S	This is an extremely useful paper with important policy implications. It became clear that half the families who experience food insecurity are not eligible for FSM. There was also a strong correlation between food insecurity and anxiety, which should not be overlooked. Below is a list of recommendations that could strengthen this manuscript: List of specific comments on aspects that could be improved: P4 L5 It would be useful if statistics of the rise could be added (if available) P4 L10 Are the data related to food insecure households from before or after covid-19? P4 L13 It would be useful to draw upon the impact on physical health. In the utero and early years there is faltering growth; delayed cognitive and behavioral development, diminished immunocompetence, vitamin A deficiency and anaemia. In school years there are higher chances to develop childhood asthma and iron deficiency, which is often associated with learning impairment and decreased productivity. In case of interest, some useful references can be found here: https://www.researchgate.net/publication/355973336_Public_Health_for_Paediatricians_Fifteen-minute_guide_to_identify_and_address_food_insecurity P5 L26 It might be useful to mention why this age group was selected, is it because younger children might not understand the questions in the survey? P5 L49 I am wondering why the categorisation of 12 ethnic groups was reduced to only 3. The experiences of Black people might not be similar to 'all other'. The 'White' category includes ethnic minorities such as Travellers or Roma. Information about this ethnic groups is likely to get lost due to the decision to have larger categories. I would suggest acknowledging this in the limitations of your study. P5 L54 Similarly, some nuances are getting lost by combining occupation to two single categories (lower or higher) P6 L38 Was any standardised measure of mental health used, or was this based on this single question? Mental health includes a whole range of aspects, anxiety or stress being only of these aspects. It would be worth it to clarify in the abstract that this is what you mean by 'mental health' P7 L24 This sentence is slightly unclear although I understand what you mean: 'a fifth of children reported any measure of food insecurity' – could you please rephrase to make this more clear?
---

	P7 L41 Not sure what 'data not shown' in the parenthesis means? P8 L38 It is recommended using the same tense when reporting past results to make the language more consistent. For example 'were unemployed' 'are persistently employed'. P9 L48 The author introduce some very insightful information here, which is that children were asked whether having FSM was embarrassing. It is recommended that this is also reported in the methods section (e.g. what children were asked) as well as in the results section. Some readers might miss that information if they don't read the whole discussion section. However, de-stigmatising this issue is key in addressing child food insecurity in the future. P10 L10 It should be also added that there are not insights as to whether some ethnic groups experience higher ethnic groups (due to the decision to categorise all ethnic groups under 3 broad categories). The same applies to occupation that had only 2 categories (ie. lower vs higher). Also, there is no information regarding non-binary groups. P15 L47 It is unclear why ethical approval was not required for a study that involved human participants. Particularly considering the sensitive nature of this subject. Some justification on this matter is needed. Table 1: The mean should be reported separately from the SD Table 1: The authors should clarify that the category 'Other' includes 'Black' and is mixed with all 'other' ethnic groups. If the data is available ideally it should be a separate category.
--	--

REVIEWER	Yau, Amy London School of Hygiene & Tropical Medicine, Public Health, Environments & Society
REVIEW RETURNED	05-Dec-2021

GENERAL COMMENTS	This paper addresses the important topic of child food insecurity during the COVID-19 pandemic in relation to free school meals (FSM), a government programme thought to be important in mitigating the negative health effects of child poverty and food insecurity. Your work highlights the unmet need among families with children, who are either not eligible for or not utilising FSM. Your work also indicates that food insecurity requires government action beyond FSM, with FSM recipients still experiencing food insecurity and the negative consequences of food insecurity. I have provided some comments and suggestions below that I hope will improve the paper. My main comments are related to details omitted from the Methods section and a suggestion to use interaction terms to look at child food insecurity, FSM and stress. METHODS  • Please include information on the response rates and final number of responses for each survey • It is unclear from the text how many participants had missing data for each variable (e.g. ethnicity, FSM) and whether these participants were dropped from the relevant analyses or treated as a separate group within analyses. • Were ethnicity and occupation self-reported? • How were the occupation categories chosen? You have designated homemaker and casual worker as lower occupation. However, it could be that high household socioeconomic position/income is the reason why one parent can be a homemaker or work infrequently. Do you have occupation data for only one parent? Do you have data on household income? Associations may be different if you looked at some measure of household-level socioeconomic position. • Can you state whether the food insecurity questions are based on a standardised tool?
--

	 • Why were the surveys combined? Would this have led to any biases (e.g. due the pandemic context at that time)? Is it possible to look at the two time points separately as a repeated cross-sectional analysis? Or perhaps an additional analysis looking longitudinally at the sub-sample who responded to both surveys (as this sample looks too small to be the main analysis)? • Would removing repeat responders from survey one bias the sample? Are responders to both surveys different from participants who responded to one survey? • How many participants were included in the analysis looking at stress? Please include this information in the text and in Table 3. • You state that “Child age and sex were additionally adjusted to improve precision.” I think that adjusting for child age and sex is important to account for potential confounding but does not improve precision. • Instead of stratifying by food insecurity or receiving FSM, you could perhaps consider using interaction terms. It seems to me that the more policy-relevant question here would be 1) what is the relationship between food insecurity and stress (which you answer), and then 2) does receiving FSM reduce the odds of food insecure children reporting stress? Rather than looking at whether FSM is separately associated with stress. RESULTS  • As single-parent households are disproportionately affected by food insecurity, it would be good to describe prevalence of single-parent households as well as overall household size. Do you have this information? • Have you considered grouping age differently, as there are only 8 participants in the 18-24 age group? GENERAL COMMENTS  • It may be useful to mention somewhere that children in your sample are outside of the age range for universal FSM • Please include a STROBE checklist • Please provide information on participant consent and ethics approval
--	---

REVIEWER	Parnham, J Imperial College London, Primary Care & Public Health
REVIEW RETURNED	21-Dec-2021

GENERAL COMMENTS	In the introduction its important to distinguish UIFSM from FSM as the figure of 1.63 million children eligible for FSM doesn't include UIFSM children.
---

REVIEWER	Stewart, Kitty The London School of Economics and Political Science Department of Social Policy, Centre for Analysis of Social Exclusion
REVIEW RETURNED	06-Jan-2022

GENERAL COMMENTS	BMJ Open (bmjopen-2021-059047) Are Free School Meals failing families? Exploring the relationship between child food insecurity, child mental health
--

and Free School Meal status during COVID-19: national cross-sectional surveys.

Referee Report, January 2022

This paper examines data from two UK national surveys conducted online in August-September 2020 and January-February 2021, to explore the prevalence of food insecurity and its relationship with Free School Meal (FSM) status. It reveals high levels of food insecurity, and shows that while children registered for FSM are more likely to be food insecure, there is still a high level of food insecurity among those not registered. The authors argue that this shows the need to extend the criteria for FSM beyond the current highly restrictive criteria.

The topic is an important one, and while the analysis is simple and descriptive, the paper provides very useful and timely evidence on the extent of food insecurity in the UK, including among children not registered for FSM, and on the correlation between food insecurity and poor mental health among children. I support publication, though I recommend a few revisions with a view to strengthening the paper further.

1. The authors argue that the sample for the surveys is geographically and demographically representative of adults living in the UK with children and young people aged 7-17. The degree of representativeness is important given the paper reaches conclusions about the share of children in the UK who are affected by food insecurity (and it is a very high share). It would be very useful to have some comparison of the sample with that in a larger scale survey, e.g. the FRS, to have a better sense of how the sample compares, and some more reflection on whether the sample may be skewed towards families who are more food insecure. The share of children reporting they are registered for FSM is much higher than reflected in national data, as the authors acknowledge, and the share using a foodbank also seems very high. The authors argue that the high FSM rate could be due to families becoming very recently eligible as a result of the pandemic (though it should now be possible to access data for January 2021 to check this). While this is one explanation, another is that the sample is biased towards more food insecure households, perhaps because these households were more interested in answering the survey. I think this merits some attention.
2. In the methods section it could be a bit clearer which questions are answered by the children and which by the parents/guardians. I think 'participants' always means children but it would be good to clarify this, as the survey

	has been introduced as a survey of parents and children, and the descriptive statistics all relate to the parents.  3. It would be good to see some descriptive statistics on the children themselves, at least age and sex. 4. A little more could be said about informed consent among the child participants. 5. Do we know how many of the households are lone parents? Could lone parenthood help to explain the surprising result that food insecurity is greater among parents with higher than lower occupational status? 6. Did the parents themselves answer any questions about food insecurity? It would be helpful to discuss this as an obvious question arises about the overlap and consistency between what parents and children say. Similarly with regard to visits to foodbanks – do children always know whether the family has been to a foodbank? Data from the parents might be interesting as a check. 7. Because there is no information about parental income or other financial resources, food insecurity is the only indicator of poverty in the dataset. This could be taken into account a bit more carefully in discussion about the association between food insecurity and poor child mental health. The implication is that it is lack of food itself which is resulting in poor mental health, but it could also reflect other aspects of living in poverty (e.g. anxiety and stress among adults in the household). There is a brief mention of this – the complex poverty-related stressors of living in a household eligible for FSM (p.8) – but the fact that the data are not very rich in regard to picking up wider household circumstances could be a bit more centred in the discussion. 8. I was confused by the interpretation of the regressions. It is argued (p.8) that because the strength of association between food insecurity and stress/worry was reduced when FSM receipt was included in the regression, this indicates that FSM helped alleviate the burden of poverty and food insecurity. I didn't follow this point. It seems to me that including FSM in the regression means we have another measure of poverty in the regression, and this soaks up some of the relationship between poverty and stress which is otherwise all being captured in the food insecurity coefficient. I don't think it tells us anything about the role FSM plays in reducing food insecurity. (I'm sure school meals do indeed play such a role, but I don't see how it can be untangled from these data.)
--	---

VERSION 1 – AUTHOR RESPONSE

Reviewer: 1

Dr. Zoe Moula, Imperial College London

Comments to the Author:

This is an extremely useful paper with important policy implications. It became clear that half the families who experience food insecurity are not eligible for FSM. There was also a strong correlation between food insecurity and anxiety, which should not be overlooked. Below is a list of recommendations that could strengthen this manuscript:

List of specific comments on aspects that could be improved:

P4 L5 It would be useful if statistics of the rise could be added (if available)

An additional sentence has now been added:

“In that decade, the Trussell Trust reported a 31-fold increase in the number of emergency food parcels distributed, from 61,000 in 2010-2011 to 1.9 million in 2019/2020 [1].”

P4 L10 Are the data related to food insecure households from before or after covid-19?

The data are from the period prior to the COVID-19 pandemic and this has been clarified in the sentence.

P4 L13 It would be useful to draw upon the impact on physical health. In the utero and early years there is faltering growth; delayed cognitive and behavioral development, diminished immunocompetence, vitamin A deficiency and anaemia. In school years there are higher chances to develop childhood asthma and iron deficiency, which is often associated with learning impairment and decreased productivity. In case of interest, some useful references can be found here:

https://www.researchgate.net/publication/355973336_Public_Health_for_Paediatricians_Fifteen-minute_guide_to_identify_and_address_food_insecurity

These are important associations with food insecurity and we have altered the sentence to highlight that there are considerable nutritional, physical, and cognitive implications which are not limited to those listed and included an additional reference (<https://doi.org/10.3390/nu13030911>).

P5 L26 It might be useful to mention why this age group was selected, is it because younger children might not understand the questions in the survey?

We have included an explanatory sentence for why this age group was selected:

“Children younger than 7 years of age, including children aged 7 in Year 2 of primary school, were excluded in order to capture children’s experiences outside of universal FSM provision.”

P5 L49 I am wondering why the categorisation of 12 ethnic groups was reduced to only 3. The experiences of Black people might not be similar to ‘all other’. The ‘White’ category includes ethnic minorities such as Travellers or Roma. Information about this ethnic groups is likely to get lost due to the decision to have larger categories. I would suggest acknowledging this in the limitations of your study.

This is an important point. We did not include the Black ethnic group as a separate category because we felt the sample size (n=82) was too small. The proportion of those with “Other White Background” included in the “White” ethnic category was relatively small (n=89). We have acknowledged this as a limitation in our study with the following additional sentence:

“We were unable to explore more granular categories of ethnicity given small sample sizes among, for example, children of Black ethnicity, preventing examination of how other prevalent ethnic groups experience food insecurity.”

P5 L54 Similarly, some nuances are getting lost by combining occupation to two single categories (lower or higher)

The collapsed occupational categories used were categories created by ChildWise to collapse the occupational categories into the National Readership Society (NRS) social grade, with the higher occupational class as shorthand for middle class (ABC1) and the lower occupational class as shorthand for working class (C2DE). This has been clarified in the *Sociodemographic characteristics* Methods section:

“Twelve categories of occupations were collapsed by ChildWise into two categories of social grade (ABC1 and C2DE), with the higher occupational class as a shorthand for middle class (ABC1) and the lower occupational class as shorthand for working class (C2DE):”

P6 L38 Was any standardised measure of mental health used, or was this based on this single question? Mental health includes a whole range of aspects, anxiety or stress being only of these aspects. It would be worth it to clarify in the abstract that this is what you mean by 'mental health'

Our measure of mental health consisted of a question asked within the survey which was not a standardized measure of mental health. This broad approach allowed children and young people from a wide range of ages to understand and respond to the question. We have clarified in our abstract that "mental health" is encapsulated by feelings of stress and worry. The modified sentence is: "Main outcome measures: Participant characteristics were described by food security and FSM status; odds of poor mental health, reported as children reporting feeling stressed or worried in the past month, by food security and FSM status, adjusted for confounding variables."

P7 L24 This sentence is slightly unclear although I understand what you mean: 'a fifth of children reported any measure of food insecurity' – could you please rephrase to make this more clear?

The beginning of the sentence has now been rephrased to:
"Over 20% of children reported food insecurity, based on positively responding to any measure,...".

P7 L41 Not sure what 'data not shown' in the parenthesis means?

This phrase has now been removed.

P8 L38 It is recommended using the same tense when reporting past results to make the language more consistent. For example 'were unemployed' 'are persistently employed'.

This has now been corrected.

P9 L48 The author introduce some very insightful information here, which is that children were asked whether having FSM was embarrassing. It is recommended that this is also reported in the methods section (e.g. what children were asked) as well as in the results section. Some readers might miss that information if they don't read the whole discussion section. However, de-stigmatising this issue is key in addressing child food insecurity in the future.

We have described this in the *Mental health* Methods section and included it in Tables 1 and 2.

P10 L10 It should be also added that there are not insights as to whether some ethnic groups experience higher ethnic groups (due to the decision to categorise all ethnic groups under 3 broad categories). The same applies to occupation that had only 2 categories (ie. lower vs higher). Also, there is no information regarding non-binary groups.

We have included additional sentences to that described in comment P5 L49:
"Similarly, categorisation of occupation into two groups may mask differences between occupations within each group and we only had occupation data for the Chief Income Earner, which may not reflect the socioeconomic position of the household. We were additionally unable to examine the experiences of children who identify outside of the male-female binary as this information was not collected."

P15 L47 It is unclear why ethical approval was not required for a study that involved human participants. Particularly considering the sensitive nature of this subject. Some justification on this matter is needed.

This has been addressed in the editor's comments and reproduced here:

"We performed a secondary analysis of data that were collected externally by ChildWise (commissioned by the Food Foundation). We have discussed the approach taken by ChildWise extensively, who explained that ethical approval is only sought if they feel the questions or topic area are of a sensitive or personal nature. ChildWise surveys have "topic disclaimers" at the start and, where appropriate, respondents are provided with a list of helplines they could reach out to, and potentially sensitive questions include a "prefer not to say" option. When the research was commissioned it was not a pre-existing requirement from the client commissioning ChildWise. Informed consent was not obtained among child participants as parental participation in the survey and their agreement that a child within

their household would complete the child's portion of the survey was taken as assent. Data from this national survey are not identifiable."

Table 1: The mean should be reported separately from the SD Table 1: The authors should clarify that the category 'Other' includes 'Black' and is mixed with all 'other' ethnic groups. If the data is available ideally it should be a separate category.

We have not reported the mean separately from the SD in Table 1 as there is only one variable row that would require this display as the other variables are presented as percentages. However, we are happy to make any necessary formatting changes based on editorial requirements. We have included a footnote to detail the ethnic groups in the "Other" ethnic category: †The Other ethnicity category includes the following groups: Black African, Black Caribbean, other Black background, mixed, and other background.

Reviewer: 2

Dr. Amy Yau, London School of Hygiene & Tropical Medicine Comments to the Author:

This paper addresses the important topic of child food insecurity during the COVID-19 pandemic in relation to free school meals (FSM), a government programme thought to be important in mitigating the negative health effects of child poverty and food insecurity. Your work highlights the unmet need among families with children, who are either not eligible for or not utilising FSM. Your work also indicates that food insecurity requires government action beyond FSM, with FSM recipients still experiencing food insecurity and the negative consequences of food insecurity. I have provided some comments and suggestions below that I hope will improve the paper. My main comments are related to details omitted from the Methods section and a suggestion to use interaction terms to look at child food insecurity, FSM and stress.

METHODS

- Please include information on the response rates and final number of responses for each survey

In the first survey, 44,344 invitations were sent out and had a 10% response rate (n=4,490). After dropouts, screen-outs, and quotas, the final sample was n=1,065. In the second survey, 28,581 invitations were sent out with a 28% response rate (n=8,183) and following dropouts, screen-outs, and quotas, the final sample was n=1,309.

This has now been included in the *Study population and survey design* Methods section with the following through editing an existing sentence [edits in bold]:

“The first (August-September 2020; **response rate: 10%**) and second surveys (January-February 2021; **response rate: 28%**) were carried out online using a carefully constructed framework to ensure a geographic and demographic representative sample of adults living in the UK with children and young people aged 7-17 years.”

- It is unclear from the text how many participants had missing data for each variable (e.g. ethnicity, FSM) and whether these participants were dropped from the relevant analyses or treated as a separate group within analyses.

Table 1 has now been updated to include the number of missing for each variable. For regression analyses, each was conducted as complete case analysis. This has been made clearer in the *Data analysis* Methods section:

“In the subset of survey questions on children’s mental health, unadjusted and adjusted logistic regression using complete case analysis were run and estimated marginal means obtained. Fully-adjusted analyses were performed with n=1,265 participants.”

- Were ethnicity and occupation self-reported?

These were self-reported and we have updated the relevant text in the Methods section to confirm this.

- How were the occupation categories chosen? You have designated homemaker and casual worker as lower occupation. However, it could be that high household socioeconomic position/income is the reason why one parent can be a homemaker or work infrequently. Do you have occupation data for only one parent? Do you have data on household income? Associations may be different if you looked at some measure of household-level socioeconomic position.

We agree that it would be beneficial to examine other measures of household-level socioeconomic position. The dataset provided by ChildWise only had these occupational categories and their contraction into the ABC1 and C2DC social grades. Occupational categories reported were for the Chief Income Earner in the household, defined as the person in the household with the largest income. If the Chief Income Earner was retired with an occupation pension, then the most recent occupation was reported, and if the Chief Income Earner was not in paid employment but has been out of work for less than 6 months, then the most recent occupation was reported. We were therefore unable to determine the overall household income or assess other measures of household-level socioeconomic position and have included this in our limitations discussion:

“Similarly, categorisation of occupation into two groups may mask differences between occupations within each group and we only had occupation data for the Chief Income Earner, which may not adequately reflect the socioeconomic position of the household.”

We have clarified that parental occupation was reported for the Chief Income Earner in the *Sociodemographic characteristics* Methods section, which has now been reworded to:

“Parental occupation was reported for the Chief Income Earner in the household, defined as the individual within the household with the largest income. If the Chief Income Earner was not in paid employment but has been out of work for fewer than 6 months, the most recent occupation was reported. If the Chief Income Earner was retired with an occupation pension, then the most recent occupation was reported. Twelve categories of occupations were collapsed into two categories of social grade (ABC1 and C2DE), with the higher occupational class as a shorthand for middle class (ABC1) and the lower occupational class as shorthand for working class (C2DE):”

- Can you state whether the food insecurity questions are based on a standardised tool?

The food insecurity questions are not based on a standardised tool and this has been noted as a limitation with the addition of this sentence:

“The questions used to assess food insecurity...were not from a standardised tool and therefore have not been validated; however, these questions still provide insight into the disruption of quality or quantity of food available....”

- Why were the surveys combined? Would this have led to any biases (e.g. due the pandemic context at that time)? Is it possible to look at the two time points separately as a repeated cross-sectional analysis? Or perhaps an additional analysis looking longitudinally at the sub-sample who responded to both surveys (as this sample looks too small to be the main analysis)?

The surveys were combined to increase the sample size for results shown in Table 2. It is possible that the stage of the pandemic experienced by the two groups may differ; we have included an additional supplementary table (Supplementary file 1) to show Table 1 stratified by survey. This showed a difference in parent occupation, with 60% of parents in the January-February 2021 survey from the higher occupation category compared to 65% from the August-September 2020 survey. There were no other differences between characteristics, including by potential food insecurity, any food bank use, and receipt of FSM suggesting it was appropriate to combine the surveys.

We have included a sentence in the *Data analysis* Methods section to indicate the addition of this table: “We examined differences in characteristics by survey period and did not find differences by measures of food insecurity or receipt of FSM (Supplementary file 1).”

We have also included a sentence in our limitations section:

“We also combined responses across two surveys and observed that there were fewer parents in the higher occupation category at the second survey (60%) compared to the first survey (65%) but did not find any differences in food insecurity, food bank use, or FSM when we examined participant characteristics by survey, suggesting it was appropriate to combine surveys.”

We are unable to conduct a repeated cross-sectional analysis given the child mental health question was only asked at the second survey. We are also unable to implement an additional analysis looking longitudinally at the sub-sample of participants who responded to both surveys. This is because the surveys were conducted in such a way that a different child in the family could have completed the child section at each survey if there were outstanding quotas for age, gender.

- Would removing repeat responders from survey one bias the sample? Are responders to both surveys different from participants who responded to one survey?

The n=206 who responded to both surveys were included in the total sample for Tables 1 and 2 and were kept in the January-February 2021 survey in order to increase sample size for the logistic regression analysis. We have included an additional table (Supplementary file 2) showing the Table 1 characteristics stratified by participants who participated in both surveys and participants who were only captured in the August-September 2020 survey. These results show that participants who responded

to both surveys were less likely to have used a food bank (16% compared to 26%) and were less likely to be food insecure (25% vs 35%). Similar findings were observed when we compared the participants who responded to both surveys with those who only responded to the January-February 2021 survey.

We have included the following in the *Data analysis* Methods section:

“Participants who responded to both surveys were less likely to have used a food bank or be food insecure (Supplementary file 2)”

“As some respondents participated in both surveys, we removed them from the August-September 2020 survey and included them in the January-February 2020 survey in order to maximize sample size for the logistic regression analyses; we examined whether participants who responded to both surveys were different from those responding to only the August-September 2020 survey and found that they were less likely to have visited a food bank or report food insecurity. Removing these participants from the analyses did not alter our results.”

- How many participants were included in the analysis looking at stress? Please include this information in the text and in Table 3.

There is a column in Table 3 indicating the number of participants included in the analysis and this has now been included in the *Data analysis* Methods section:

“Fully-adjusted analyses were performed with n=1,265 participants.”

- You state that “Child age and sex were additionally adjusted to improve precision.” I think that adjusting for child age and sex is important to account for potential confounding but does not improve precision.

We used this language as child age and child sex in our DAG were associated with our outcome of interest but not with our exposure of interest. We have removed this and the sentence has been modified to state the inclusion of child age and sex in the adjusted models.

- Instead of stratifying by food insecurity or receiving FSM, you could perhaps consider using interaction terms. It seems to me that the more policy-relevant question here would be 1) what is the relationship between food insecurity and stress (which you answer), and then 2) does receiving FSM reduce the odds of food insecure children reporting stress? Rather than looking at whether FSM is separately associated with stress.

Thank you for this suggestion. We have modified our analyses to include a model examining the relationship between food insecurity and child report of stress/worries and an additional model with the inclusion of an interaction term between the variables for food insecurity and FSM with our outcome of child-reported stress/worries. We report our results using marginal means to aid in our interpretation.

RESULTS

- As single-parent households are disproportionately affected by food insecurity, it would be good to describe prevalence of single-parent households as well as overall household size. Do you have this information?

Parents were asked to report the number living in the household including themselves. A value of “2” can be assumed to mean that this is a single-parent household. The prevalence of this is listed in Table 1 under the heading “Number in household” and this is at 7.4%.

- Have you considered grouping age differently, as there are only 8 participants in the 18-24 age group?

We chose to use these age categorizations, created by ChildWise, in order to assess whether there were young parents and whether their children were experiencing food insecurity or receiving FSM.

GENERAL COMMENTS

- It may be useful to mention somewhere that children in your sample are outside of the age range for universal FSM

This is an important point that was also picked up by another reviewer. An additional sentence was included in the *Study population and survey design* section:

“Children younger than 7 years of age, including children aged 7 in Year 2 of primary school, were excluded in order to capture children’s experiences outside of universal FSM provision.”

- Please include a STROBE checklist

This has now been done.

- Please provide information on participant consent and ethics approval

This has been addressed in the editor’s comments and reproduced here:

“We performed a secondary analysis of data that were collected externally by ChildWise (commissioned by the Food Foundation). We have discussed the approach taken by ChildWise extensively, who explained that ethical approval is only sought if they feel the questions or topic area are of a sensitive or personal nature. ChildWise surveys have “topic disclaimers” at the start and, where appropriate, respondents are provided with a list of helplines they could reach out to, and potentially sensitive questions include a “prefer not to say” option. When the research was commissioned it was not a pre-existing requirement from the client commissioning ChildWise. Informed consent was not obtained among child participants as parental participation in the survey and their agreement that a child within their household would complete the child’s portion of the survey was taken as assent. Data from this national survey are not identifiable.”

VERSION 2 – REVIEW

REVIEWER	Moula, Zoe Imperial College London, Public Health
REVIEW RETURNED	22-Mar-2022

GENERAL COMMENTS	Many thanks for making major revisions to this manuscript. This reads very well now and this version is significantly stronger than the previous one submitted. I really believe this article will be valuable and well done for working so hard on this.
---

REVIEWER	Yau, Amy London School of Hygiene & Tropical Medicine, Public Health, Environments & Society
REVIEW RETURNED	31-Mar-2022

GENERAL COMMENTS	The authors have adequately addressed all of my comments. I am now happy to recommend publication.
--